# Do Eco-Based Adaptation Measures Enhance Ecosystem Adaptation Services? Economic Evidence from a Study of Hillside Forests in a Fragile Watershed in Northeastern Taiwan

**Wan-Jiun Chen** [1] , **Jihn-Fa Jan** [2] , **Chih-Hsin Chung** [3] **and Shyue-Cherng Liaw** [4],*

1   Department of Economics, Chinese Culture University, No. 55, HawKang Rd., Taipei 111, Taiwan; cwj@ulive.pccu.edu.tw or chenwanjiun@gmail.com
2   Department of Land Economics, National Chengchi University, No. 64, Section 2, Zhinan Rd., Taipei 116, Taiwan; jfjan@nccu.edu.tw
3   Department of Forestry and Natural Resources, National Ilan University, No. 1, Section 1, Shennong Rd., Yilan City 260, Taiwan; chchung@ems.niu.edu.tw
4   Department of Geography, National Taiwan Normal University, No. 162, Section 1, Heping E. Rd., Taipei 106, Taiwan
*   Correspondence: liaw@ntnu.edu.tw; Tel.: +886-7749-1649

**Abstract:** As the risks of climate change keep increasing, countries have emphasized the ecosystem adaptation policy, and the United Nation Environmental Program (UNEP) aids countries to adapt to a warming world with eco-based adaptation (EbA) measures for good ecosystem governance for boosting ecosystem adaptation services (EAS). With the purpose of helping to indicate the magnitude of the benefits of EAS from local EbA measures, this study assesses the economic value of the EAS of hillside forests regarding the residents in a climate vulnerable watershed, the Lanyang River watershed, by applying a single-bounded contingent evaluation method. The demographic variables and motivation variables indexed by perceived impacts are influencing factors in the residents' willingness-to-pay. These variables are of significance in EbA policy application. The average economic value for each responding resident was estimated to be NT$ 793.65 on the basis of a survey of the residents' willingness to pay for EAS and the single-boundary contingent valuation method. The results verified that the residents depend on the protection of natural hillside ecosystems. Considering the complex interactions between ecosystems and humans, the EbA is demonstrated to be a crucial method for mitigating the consequences of climate change. Protecting hillside ecosystems in the Lanyang River watershed through soil and water management presents critical policy implications. Now that climate change has become an emergency, this case study shows the success of Taiwan's long manipulated EbA for EAS, with evidence of residents benefiting. This Taiwan case study has policy implications for the world and UNEP's global EbA program to maintain EAS.

**Keywords:** watershed protection; ecosystem assessment; ecosystem-based adaptation; ecosystem services; contingent valuation method

## 1. Introduction

Due to its rapid development, humankind needs to think deeply on our relationship with nature [1]. It is urgent for us to start from the concept of ecological protection and explore ecological solutions. The precious buffers for humankind's future survival, development, and adaptation to global change are shrinking rapidly. In the 2019 Special Report on Climate Change and Land [2], the Intergovernmental Panel on Climate Change of the United Nations stated that land degradation affects more than a quarter of the world's land. In the face of intensifying climate change, ensuring the protection and restoration of ecosystems constitutes a key solution to the climate crisis. The governance of the ecosystem mechanisms that buffer the effects of climate change is a prominent concept associated

with eco-based adaptation (EbA, hereafter). EbAs are climate adaptation measures that strategically maintain and emerge ecosystem services to climate adaptation by increasing the capability of ecosystems to moderate and adapt to climate change and variability. The congregations of ecosystem services that enable human adapt to climate change are named Ecosystem Adaptation Services (EAS, hereafter).

Since the ongoing shifts in weather patterns triggered by climate change have altered rainfall and temperatures and affected the ecosystem services required for human life [3], the measure of EbA is set for increasing climate resilience by enabling the power of natural ecosystems and enabling humans to adapt to the effects of climate change [4–6].

EbA practices can be integral to climate adaptation strategies at the global, national, and local levels [5,7–9], through scientifically and politically implementing market-based and technology-based techniques that coordinating natural and social system [10,11]. The local ecosystems may vary widely across communities. Based on local landscape and ecological characteristics, EbA could be a viable nature-based approach that adopt locally operationalizable practices to harness ecosystem services to buffer communities from climate change [12–15]. Such practices would fulfill ecosystem services including adjusting and cooling microclimates, enhancing surface water infiltration, reducing runoff, replenishing groundwater, protecting slopes and water banks, and improving water and soil conservation. The empirical practices of EbA enhance climate ecosystem services and benefit residents. The benefit covers wide ranges of ecosystem services [16]. A quantitative evaluation of the economic benefits of EAS can demonstrate the value of natural systems in local climate adaptation.

EbA is associated with numerous long-standing strategies in the fields of natural conservation and community-based natural resource management. It can be enhanced with technical and engineering solutions. EbA strategies are widely applicable and meaningful to vulnerable communities, particularly those comprising citizens that depend on the environment for their livelihood, since the "United Nations Framework Convention on Climate Change" had compiled information of ecosystem-based approaches to adaptation [17].

Developing local community adaptation policies to conserve and invest in natural ecosystems can enable coordination of dynamic services delivered from healthy ecosystems. Although the importance of EbA has gained recognition, few studies have provided empirical evidence of its economic contribution. Implementing agencies must balance costs and benefits to conserve precious time and resources.

Professor William Nordhaus of Yale University, a Nobel Laureate, regards the climate crisis as the most urgent threat to humanity's health and happiness and evaluates costs and benefits using aggregate economic analysis to establish basic principles for rational decision-making [18]. The aggregate economic analysis approach can be employed to clarify aggregate questions and identify scientific, economic, and policy issues. Aggregate climate economics can be used to simplify the fundamental concepts of economic values. In addition to the aggregate approach, employing downscaling techniques can yield detailed information on climate change at the local level. An in-depth case study would effectively address heterogeneity in local ecosystems. Estimating the economic value of EAS to reduce the impacts of climate change at the community level is crucial for the successful transition through climate change. EbA is closely related to local land classification, land-use planning [19], land consolidation [20], and natural conservation planning and management [15,21]. Ecosystem services play important roles in improving the quality of life and socioeconomic opportunities of the vulnerable residents [22]. EbA would be a promised vehicle for social empowerment to local climate adaptation [23]. As regard, United Nation Environmental Program (UNEP) aids countries to adapt to a warming world with EbA measures to good ecosystem governance for boosting EAS.

This paper aims to assess the benefit of EbA, by taking the example case of Lanyang River watershed in Northeastern Taiwan, a vulnerable hillside forest protected by EbA practices. EbA measures are particularly important for the Taiwan and Lanyang River watershed due to the climate patterns and steep mountainous topography. The paper

evaluates the resident for benefit of EAS that maintained and emerged by EbA practices by using the single-bounded contingent evaluation method.

Taiwan is an island located in Southeastern Asia. The island covers an area of 36,197 km, with an altitude of 3452 m. The mountains in Taiwan are steep, the coastal plains are small, and the population is dense. Lanyang River watershed is located northeastern Taiwan, and covers small plains and high mountainous areas. This basin is rich in biodiversity and is a crucial area for agriculture and outdoor recreation in Taiwan. However, the watershed is one of the most vulnerable areas in Taiwan. The climate of Taiwan ranges from tropical to subtropical. Uneven rainfall in the monsoon and typhoon seasons poses a particular challenge to Taiwan's slope land areas. Extreme weather events often cause substantial damage in Taiwan [24,25].

Lanyang River watershed is on the windward side of Typhoon. The watershed frequently experiences strong winds, torrential rains, and typhoons, which adversely affect the ecosystem and environment. Extreme weather events, typhoons, and torrential rain often lead to the loss of soil and water resources and the destruction of sloping animal and plant habitats, posing a potential threat to the ecosystem services. The local area is highly vulnerable and often affected by serious threats to the safety of residents' lives and property, including floods, landslides, and mudslides. Since these impacts are serious threats to their life and property, and the residents suffer since their early access to land reclamation and settlement [24,25].

Due to Taiwan's topography of high mountains and steep slopes, Taiwan's government' self-financed EbA measures had long been a sound and prominent policy that requisite to Taiwan's development [24]. Especially, the Lanyang River watershed in northeastern Taiwan is a climate vulnerable zone. Local residents are the primary interest group influenced by ecosystem services to adapt climate change. The EbA practices would vary with the local specific ecosystem characteristics. In Taiwan, land zoning, hillside forest conservation, and soil and water conservation practices [24] in hillside forested areas are the main tasks of local EbA measures that enable EAS to serve the resident.

The varied topography due to a large altitude difference in the watershed affects its ecosystem services. Since the land use in Taiwan was classified and zoned, the hillside of this watershed is the frontiers of natural ecology and human society. From the perspective of climate adaptation, the effective maintenance of hillside ecosystems can play a considerable role at the socioeconomic level.

Under the increasing risks of climate change, and efforts to reduce carbon emissions not enough to mitigate climate change impacts, the UNEP aids countries to adapt to a warming world with eco-based adaptation (EbA) measures to good ecosystem governance for boosting EAS.

With the high risk of climate change, the Lanyang River watershed encounters challenges, especially at the hillsides and the development front [24,25]. Assessing the economic value of EAS can generate an indicator to affirm the requisite and the success of EbA measures. In addition to assessing the economic value of hillside forests' EAS in Lanyang River watershed, the policy implications would be made for the world and UNEP's global EbA program to maintain EAS. This study would apply the findings to confirm the importance of nature-based solutions and EbA measures. Economic evidence can support the delivery of nature-based solutions to ecosystem service challenges to reduce vulnerability to climate change. This study seeks to provide economic evidence that EbA practices can enhance climate resilience by harnessing the power of natural ecosystems to benefit the people who depend on them.

It is to be noted that the same study site was investigated for a wider range of ecosystem services by Chen et al. [25]. Chen et al. studied the benefit of the local residents in terms of their willingness to pay for ecosystem services in hillside forests with respect to four categories of ecosystem services that identified by Millennium Ecosystem Assessment (MEA, hereafter) [26] in their open-access report. The two economic evaluation studies are performed with different surveys, different questionnaires, and screening different

resident samples. Together with the present study that investigate the residents' benefit of the EAS, ecosystem services can illustrate and expand recognition of the benefits of ecosystem services and their linkages with human society.

## 2. Materials and Methods

### 2.1. Ecosystem Services and Climate Adaptation

The dynamic relationship between ecosystems, ecosystem services, and climate adaptation is extremely complex [27]. The governmental and scientific attentions have paid to build a harmonious relationship between human society and river ecosystems [28]. Land-use change would change ecosystem services, and affects the use of ecosystem services to adapt to the challenges of local, regional, and global climate change. The hillside is the frontiers of natural ecology and human society. A land-use change would affect ecosystem services [29]. Good governance of the ecosystems would maintain our life support systems [30]. The governance of ecosystem and ecosystem services has emerged as a prominent policy measure for adaptation [31]. Conventional measures involving restoring ecosystems on the basis of historical information are impractical. Harris et al. [31] posited that a more practical approach is protecting at-risk ecosystems and maintaining ecosystem services. Through this approach, natural capital can be sustained, which enhances its provision of services locally, regionally, and globally.

Humanity faces the reality that human development and climate change are contributing to environmental change at an unprecedented pace. Climate change is among the most urgent challenges faced by humans [32]. The past no longer foreshadows what may occur in the future. Ecosystem restoration requires intensive, long-term efforts. Rapid climate change may cause the public to grow impatient with and lose interest in the goal of ecosystem restoration [31]. To adapt to such changes, a resilient natural system must be established for current and future generations. Feasible climate measures should be developed on the basis of the current environmental status. The forces that guide the transition of socioeconomic systems should be systematically guided to coevolve with nature [33–35]. EbA can be applied in remote, rural, and urban areas, as well as developing countries [36–38] and developed countries. In the relevant literature, Wamsler and Pauleit [39] present the EbA practices in 12 municipalities in Germany and Sweden.

Ecosystem services are the benefits conferred to human beings by ecosystems. In accordance with the concepts of anthropocentrism, Perrings et al. [40], Crossman et al. [41], and Lavorel et al. [42] indicated that the benefits of ecosystem services are the prominent forces guiding the policies of natural resource management. The MEA [26] identified four major categories of ecosystem services: provisioning, regulating, cultural, and supporting services. Material provisioning and cultural services directly and immediately contribute to human wellbeing, and they are reinforced by regulating and supporting services [43].

As climate change has become a policy focus, attention has shifted to the benefits of ecosystem services in improving climate change adaptation [42,44]. Lavorel et al. [42] defined climate adaptation services as the benefits humans receive because of the capability of ecosystems to moderate and adapt to climate change. Lavorel et al. [42] investigated the ecological mechanisms by which four Australian ecosystems adapt to climate change; their results demonstrated that the components for severe climate change coping in the four sites were vegetation structural diversity, keystone species, and landscape connectivity. Lavorel et al. [42] further indicated that a better understanding of the benefit of climate adaptation services can guide ecosystem management in adaptation planning.

### 2.2. EbA and Benefit Assessment

EbA is a governance and planning method for the adaptation of ecosystems to climate change. Zölch et al. [45] identified ecosystems, ecosystem services, and EbA as three major components for the development of climate adaptation plans and strategies.

The natural solutions that were initially proposed to address environmental problems by mediating the link between science, policy, and practices [46,47] can be integrated and

extended to enable adaptation to climate change [46,48]. EbA practices can help reduce or avoid impacts of climate changes. Promoting ecology can simultaneously promote adoption [49]. EbA to climate change is a feasible measure with a scientific basis in conservation [50]. Donatti et al. [51] searchrd for effective indicators to signify the outcome of EbA.

EbA is a nature-based solution that harnesses biodiversity and ecosystem services to reduce vulnerability and strengthen resilience to climate change [7–9,15]. Ecosystem governance can benefit from climate adaptation. The dynamic mechanisms of the natural services in a healthy ecosystem can enhance resiliency to climate change. EbA is a pragmatic approach to buffering the effects of climate change [52]. Ecosystem services have been widely harnessed as an adaptation strategy [15,52].

With complexity in the dynamic interactions between human society and ecosystem services in a watershed, establishing an appropriate value index can directly guide the watershed management and serve as the basis for a feasible measure of climate adaptation [53]. We need an indicator to demonstrate the outcome services achieved. As indicated in the literature [46,54,55], the estimating for the monetized amount of economic value could generate the effective indicators and knowledge, which can be employed by the public to create actions and seize opportunities to adaptation.

Although EbA has gained academic recognition, empirical evidence on its economic value for associated interest groups remains weak. The present case study of the Lanyang River watershed in Taiwan investigated local residents' willingness to pay for the perceived benefits of EbA practices with respect to climate change by using the contingent valuation method. This watershed is one of the most vulnerable in Taiwan, and the present study's evaluation is crucially consequential for its ecological protection and for the implementation of EbA policies. Furthermore, this research may enable the promotion of relevant policies.

EbA measures can be facilitated using climate governance planning [45]. Our study evaluated the benefits of hillside EAS in the Lanyang River watershed. The results can provide a monetary indicator of the benefits of EbA.

In the literature, studies had addressed opportunities for mainstreaming EbA into the climate agenda [15,52,56–58]. (1) EbA is a climate adaptation strategy that directly links to the surrounding environment. (2) EbA is a cost-effective policy that can buffer the effects of climate change. (3) EbA enables the development and customization of adaptation practices to suit local needs. (4) EbA can supplement insufficient investment in expensive infrastructure and effectively protect residents who are vulnerable to climate change. (5) Strengthening and protecting ecosystems through EbA practices is linked to long-term investments in climate adaptation. (6) EbA policies that coordinate, mediate, and harness the forces of nature are economically viable.

The aforementioned aspects of EbA indicate it is a flexible, cost-effective, and widely applicable climate adaptation measure that can effectively reduce the effects of climate change. EbA is a crucial strategy through which climate adaptation policymakers can address the threats posed by climate change to the lives and livelihoods of people worldwide [11,52,57]. The EbA would be a viable measures after overcoming constrains and challenges [57,59,60].

EbA has been adopted by numerous nations. The threat that climate change poses to the lives and survival of millions has not abated. Humanity has a shared responsibility and obligation to understand and adapt to climate change on a global scale. Planning and implementing local EbA-related policies is crucial for combating the threat that climate change poses. Vignola et al. [61] addressed the application of EbA in developing countries. The climate change adaptation policies should take full account of the role of ecosystem services in enhancing social resilience especially where economies and livelihoods are largely dependent on ecosystem services. Muthee et al. [62] reported that 168 climate adaptation policy initiatives in 13 West African countries were closely linked to the four categories of ecosystem services (i.e., provisioning, regulating, cultural, and supporting services). According to the analysis of Muthee et al. [62], 32% of these adaptation policies

in West Africa were implemented in the agricultural sector. Furthermore, 55% directly referred to one or more ecosystem service, and 50% referred to ecological provisioning services. However, a consistent policy implication emerges in developed countries, as well. Germany integrated EbA into urban climate adaptation strategies [45].

According to the relevant literature on EbA, ecosystem services are crucial to climate adaptation. Informative messages regarding the benefits of these services can guide policies related to ecosystem restoration and conservation to provide adaptation services to the general public. The present case study assesses the benefits of hillside EAS in the Lanyang River watershed to establish a monetary value for specific EAS.

### 2.3. Application of Contingent Valuation Method in Ecosystem Services Assessment

To address the challenges of climate adaptation, Seddon et al. [54] indicated the value of ecosystems, their services, and the adaptation of these services must be understood. Ecosystem services broadly benefit humans by providing a wide range of market-traded and non-market-traded goods and services. The contingent valuation method constitutes a viable method for valuing tangible and intangible commodities. Zhang and Zhao [63] (2007) analyzed the validity and reliability of the contingent valuation method in assessing the value of ecosystem services and provided guidance on research and survey designing.

Studies have widely employed the contingent valuation method to assess the value of ecosystem services. Navrud and Strand [64] applied this method to value the ecosystem services of the Amazon rainforest. Jobstvogt et al. [65] applied the contingent valuation method to value the ecosystem services of the protected areas in Great Britain from a cultural perspective. Kalfas et al. [66] assessed the benefits of the ecosystem services of urban forests. Xu et al. [67] estimated the value of wetland ecosystem services.

The contingent valuation method, which can yield valuable information for empirical applications, is often adopted in policy assessments. Müller et al. [68] employed this method to analyze the role of forest ecosystem services, and their assessment was applied in forest management practices. Wang et al. [69] analyzed the value of ecosystem services lost because of land reclamation and cultivation. According to the literature, the contingent valuation method constitutes a sound method for assessing climate adaptation policies [70].

Innovative application of new technologies can greatly enhance climate adaptation capability. To mitigate the effects of climate change, governments have implemented climate adaptation projects for water and soil governance, such as those to control irrigation, flooding, and erosion, at varying scales. Al-Amin et al. [71] employed the contingent evaluation method to analyze the willingness to pay of northern Malaysian people for the installation of adaptation technology. Their research also analyzed the demographic characteristics and behavioral drivers of the interest group.

Banna et al. [72] and Masud et al. [73] have assessed Malaysian farmers' willingness to pay for effective agricultural adaptation plans in response to climate change. Banna et al. [72] distributed a structured questionnaire to farmers in Selangor, Malaysia, and they implemented the contingent valuation method to assess the farmers' preferences and willingness to pay for climate adaptation programs. Most respondents (74%) expressed a willingness to pay for the proposed adaptation programs, and their willingness was influenced by demographic characteristics and motivational factors. In addition to farmers, the Malaysian government, social agencies, banks, nongovernmental organizations, the media, and private organizations can cooperate to raise funds for climate adaptation programs. Banking institutions, private companies, governments, and farmers can also donate to climate adaptation programs to fulfill their corporate social responsibilities.

### 2.4. Survey, Sampling, and Regression Model

The benefits of EAS include both marketable and nonmarketable services. The market system does not value the intangible benefits of natural services. The residents of the watershed are the primary stakeholders. This study assesses the benefits to residents using a questionnaire. Because the literature has demonstrated that respondents' demo-

graphic characteristics and payment motivational factors are the main determinants of their willingness to pay [71,72], the questionnaire included these determinants.

In addition, in the questionnaire, the sampled residents are asked their willingness to pay for the EbA practices that offer EAS. In this part, the respondents are firstly asked to imagine a reliable environmental foundation that is hypothetically established in the Lanyang River watershed to harness local ecosystem services to help people adapt to the effects of climate change. The foundation can be used effectively in financing the EbA adaptation practices, such as conservation, sustainable management, and restoration of ecosystems. In addition, the respondents are then asked about their willingness-to-pay with assigned bidding values to enable the foundation to implement the aforementioned policies for climate resilience and adaptation in the hillside ecosystem.

The demographic characteristics in the questionnaire are designed as 7 corresponding questions, including: (1) gender, (2) age, (3) education level, (4) occupational status, (5) household annual income, (6) if ever participating as environmental protection volunteers, (7) if ever donate for environmental protection, and (8) township of residence.

The payment motivations consisting of local climate security are measured by perceived climate impacts by Likert five scale. The 7 perceived climate impacts questions (latter being transformed as security motivation variables) included in the questionnaire are as follows.

1. I think the local temperature has gradually increased in the past ten years.
2. I think the number of high-intensity rainfall has increased in the past ten years.
3. I think the number of affected typhoons has increased in the past ten years,
4. I think the number of natural disasters (floods, landslides) has increased in the past ten years.
5. I think climate change is a very serious global problem.
6. I think climate change will affect human life.
7. I think climate change will impact the environment and ecology.

The questionnaire is designed to have four parts. It is start with the first part: a brief introduction to the purpose of the survey. The second part is the 7 perceived climate impacts measured by Likert five scale. Then, part 3 is the questions to capture the willingness to pay for the assigned bidding value. Finally, part 4 is composed of 7 demographic questions.

This study distributed the survey from August to September, 2021. Convenience sampling of residents at local railway stations was employed. In addition, during in-person interviews, the sampled residents were asked their willingness to pay a specified bidding amount. The bidding amounts (BID) of NT\$300, NT\$400, NT\$500, NT\$600, NT\$800, NT\$1000, NT\$1200, NT\$1500, NT\$2000, and NT\$3000 were randomly assigned to the interviewed respondents. Each participant indicated their willingness to pay the random specified bidding value in the questionnaire. The respondents were asked the following question in the interview.

> *Imagine that a reliable environmental foundation is established in the Lanyang River watershed to harness ecosystem services to help people adapt to the effects of climate change through the conservation, sustainable management, and restoration of ecosystems. Are you willing to pay (NT\$) <u>BID</u> per year to enable the foundation to implement the aforementioned policies for climate resilience and adaptation in the hillside ecosystem?*

> ☐ *Yes, I am willing to.*

> ☐ *No, I am not willing to.*

Among the 423 interviewed residents, 419 completed the questionnaire.

This study implemented the single-bounded dichotomous contingent valuation method. The regression function for the logit model can be expressed as follows:

$$P(Y_i) = f(BID_i, X_i) + \mu_i \tag{1}$$

where the dependent variable P(Y) is the binary response of the residents (yes = 1, no = 0); BID is the bidding value; X is a vector of dependent variables, including the respondents' demographic characteristics and motivation questions according to Al-Amin et al. [71]; $\mu$ is the regression error term; and a subscript i represents the i-th respondent. A logit model was employed to estimate the parameters. Cameron [74] indicated that the point estimates of the willingness to pay in the single-bounded contingent valuation method can be approximated using the following formula:

$$E(WTP) = -1/\hat{\beta}_1 \tag{2}$$

where $\hat{\beta}_1$ is the estimator of the bidding variable in the logit model (Equation (1)).

According to the estimated parameter of bidding variables, the expected value of willingness to pay (WTP) can be calculated. This expected WTP is estimated to represent the benefit of EAS to the resident as an indicator of economic value of EbA measures.

### 2.5. Definition of Variables and Their Associated Statistics

After the data obtained from the questionnaire, the regression considers the 7 demographic characteristics and the 7 perceived motivations as candidate explanatory variables introducing into X vectors. However, the insignificant variables are dropped out by econometric interpolative and extrapolative process. Variables in X vector were screened, only the significant variables in the logit regression model were retained. The significant variables that affected the binary dependent variable, P(Y) in Equation (1) included BID, 4 demographic characteristics, and 1 motivational factor. After screening by statistical significance, the X vector in regression equation 1 encompasses variables of perceived impacts of climate change on human life, age, education level, housewife as occupational status, and income. Table 1 presents the definitions of the variables and their associated descriptive statistics.

**Table 1.** Definitions of Variables and Their Associated Statistics.

| Variable | Definition | Mean | S.D. |
|---|---|---|---|
| Y | Dependent variable, Binary variable. =1, if the response is "yes, he/she is willing to pay the bidding amount"; =0, otherwise. | 0.49 | 0.50 |
| BID | Bidding amount for the willingness-to-pay (NTD, per year). The bidding amount asking the interviewees according to the order of sampling: 300, 400, 500, 600, 800, 1000, 1200, 1500, 2000, and 3000 NTD. | 1124.58 | 798.32 |
| LIFE_IMPACT | The perceived climate impact to live, measured by Likert five-scale, strongly agree = 5, agree = 4, neutral = 3, disagree = 2, and strongly disagree = 1. | 4.32 | 0.64 |
| AGE_31to60 | Dummy variable. =1, the respondent aged between 31 to 60 years; =0, otherwise. | 0.54 | 0.50 |
| EDU | Years of education received. | 14.13 | 3.08 |
| HW | Dummy variable. =1, if the respondent is a housewife; =0, otherwise. | 0.08 | 0.28 |
| INC | Annual household income (measured in NT$10,000) | 79.18 | 45.03 |

Regarding the motivation variables, only perceived impacts of climate change on human life (transform from the motivation question numbered 6 in the questionnaire) is retained due to its significance. The perceived impacts of climate change on human life (LIFE_IMPACT) was measured using a 5-point Likert scale (strongly agree = 5, agree = 4, neutral = 3, disagree = 2, and strongly disagree = 1). The other six motivation variables, which reflect the climate security, are dropped due to insignificance in the logistic regression.

Many demographic variables transformed from the questionnaire survey are significant. The dummy variable AGE_31to60 was employed for age; for the respondents between 31 and 60 years of age, AGE_31to60 = 1, for all other respondents, AGE_31to60 = 0. The

variable EDU represented the years of education received. The dummy variable HW was assigned a value of 1 if the respondent was a housewife and a value of 0 if not. The variable INC represented annual household income measured in NT$10,000.

## 3. Results and Discussions

The Lanyang River watershed is one of the most ecologically vulnerable in Taiwan. This case study evaluated the economic benefits of hillside EAS in this watershed related to mitigating climate change for residents. The residents' willingness to pay was estimated using the contingent valuation method. This research provides valuable insights for watershed governance for water and soil protection and for the implementation of EbA policies. The findings of this study demonstrate that EbA policies are beneficial and should be adopted in other communities in accordance with the unique traits of their local ecosystems.

### 3.1. Result of Logit Regression Estimation Results and the Influencing Factors

The estimated results of the logit model for equation 1 are presented in Table 2. The results demonstrated that a decrease in willingness to pay was associated with a higher offered bidding value. The positive signs of the variables in Table 2 revealed that the respondents with a willingness to pay the offered bidding amount tended to be (1) those with higher education levels, (2) those with higher incomes, and (3) those who perceived the impacts of climate change on their lives to be higher. Housewives and those between 31 and 60 years of age also tended to have a higher willingness to pay.

**Table 2.** Estimation Results of Logit Model.

| Variable | Coefficient | (Std. Error) | Prob. | |
|---|---|---|---|---|
| C | −4.7293 | (1.0691) | ≦0.0001 | *** |
| BID | −0.0013 | (0.0002) | ≦0.0001 | *** |
| LIFE_IMPACT | 0.4991 | (0.1828) | 0.0063 | *** |
| AGE_31to60 | 0.6063 | (0.2334) | 0.0094 | *** |
| EDU | 0.1634 | (0.0452) | 0.0003 | *** |
| HW | 0.9983 | (0.4423) | 0.0240 | ** |
| INC | 0.0145 | (0.0029) | ≦0.0001 | *** |
| McFadden R-squared | 0.2253 | Mean dependent var | | 0.4893 |
| S.D. dependent var | 0.5005 | S.E. of regression | | 0.4275 |
| Akaike info criterion | 1.1070 | Sum squared resid | | 75.2906 |
| Schwarz criterion | 1.1744 | Log likelihood | | −224.9120 |

Note: ** and *** represents significance at 5%, and 1%, respectively.

The study results consistent with the arguments for the influencing factors for the economic value of ecosystem services in the literature. According to Al-Amin et al. [71], the demographic characteristics and motivation would affect the willingness-to-pay in an ecosystem services evaluation. The arguments are affirmed by the regression results in the present study. The empirical evidence of the critical factors to biotic climate adaptation measures are presented as following.

(1) Housewives and residents aged 30–60 are of higher willingness to pay, and they would potentially play an important role in the EbA biotic adaptation measures.
(2) With high risk of extremely climate severity, life and property of the residents in the study site is seriously threatened by potential disasters, floods, slope collapses, landslides, and other impacts. The study result revealed that perceived impacts of climate change to life would be an important motivation factor to the payment willingness. The residents suffer, due to their early access to land reclamation in the watershed and latter settlement in the watershed. The inhabitants are under

threat and their awareness of the impacts would increase their payment willingness to support local EbA measures.

(3) It is worthwhile to note that Taiwan has a successful literacy policy. An average respondent has received 14.13 years of schooling (Table 1). The residents with higher education level would have higher tendency to pay for supporting the foundation. The result evidenced that having people possessing higher education is able to make the interest group having relevant knowledge is also a critical society factor in climate policy implementation. It would imply the importance of education in mediating a biotic climate adaptation measure to the society.

(4) Moreover, residents with higher income have higher willingness to pay. An income increase would enhance their affordability for the payment to support the hypothetical foundation to adapt climate change. Public income would have some potential connection to the willingness to pay to facilitate self-financed adaptation measures. It is suggested that that government financing would be demanded to implement EbA as a response to the climate emergency in low-income areas.

*3.2. Economic Value to the Resident*

This study assessed the benefit of EAS to the resident as an indicator of economic value of EbA measures.

According to Equation (2), the estimated economic value, the expected value of willingness to pay (WTP) is $E(WTP) = -\frac{1}{\hat{\beta}_1} = \frac{1}{0.00126} = 793.65$ (NT$) for each respondent.

The average resident is willing to pay NT$ 793.65 to the hypothetical foundation to support the good governance of hillside for the ecosystem services to help people adapt to the effects of climate change. They were witnesses and potential sufferers of disasters, the floods, slope collapses, landslides, and other impacts that seriously threaten their life and property, if they have no protection by the EbA. This assessment offers quantitative indication evidence that reflects the resident perceived benefit from the prudent and pragmatic practices for nature ecology protection that has long established.

Even though the watershed under the present investigation is climate vulnerable, the local ecosystem is well governed and the EAS is protected to serve residents. The local empirical EbA practices have long been developed by institutional establishment, land use classification and planning, well-enforced by legislation regulations, and by excellent public education to improve the knowledge of disaster prevention [24]. The benefit of EAS is a critical and valuable information to climate adaptation as the increasing threats from climate changes accelerate. The present study can enhance and improve the governance of ecosystem services.

It is worth to note that this assessment is a one-time evaluation. The numeric estimated value demonstrated the annual willingness-to-pay that reflects the current screened resident's personal perceived economic value, regarding the stated nonmarketable commodity. The dynamic effects over time of the EbA are not presented, as the practices continue to produce protection.

*3.3. The Need for an Economic Indicator*

The present economic valuation analysis reveals the benefit of EAS related to adaptation to climate change. This study analyzed the direct benefit of the ecosystem services to the residents from the perspective of climate adaptation and on the basis of the consumer theory of neoclassical economics. Debates on the applications and procedures of the contingent valuation method have addressed its theoretical, behavioral, and political aspects [75]. However, the technique of nonmarket valuation methods is deemed as a valid method to indicate value of ecosystem services [63]. The benefit estimated in this research generated an indicator for the benefit created from the ecosystem services to adaptation for the residents. The present study results indicate that ecosystems contribute substantially to climate adaptation services. The data verify that maintaining and promoting healthy ecosystems to ensure the fulfillment of climate adaptation services constitutes a vital climate policy.

### 3.4. Warning Indicator

EbA is the use of ecosystem services to help adaptation to climate change. The climate measure of EbA provides adaptation benefits. Estimations of economic value can also reveal the potential loss of value that can result from poor natural governance. As climate change intensifies, the natural resilience of ecosystems must be strengthened to maintain the availability of ecosystem services to local populations. Enhanced protection policies can prevent devastating losses of life and property. Conservation and protection policies are required to coordinate and balance the needs of nature and society.

Establishing a balance between the natural system and the climate adjustment mechanism of ecosystems and their services can enhance the wellbeing of residents, as evidenced by this case study on the hillside ecosystem of the Lanyang River watershed.

Hillside forest ecosystems are not only directly affected by climate change, but also play a crucial role in mitigating and adapting to climate change. The results demonstrating the dependence of the residents on ecosystem services hold essential policy implications. This vulnerable hillside ecosystem requires effective maintenance through conservation and protection policies. Ecosystem governance to protect the water and soil in the watershed can reduce the effects of climate change.

### 3.5. Ecovillages as an Alternative to Destructive Development

The purpose of this assessment research is for ecological security rather than promoting the monetization of natural assets for trading purposes.

Rapid development and intensive land use have greatly changed the land cover, damaged the structure of ecosystem and reduced the value of ecosystem services [76,77]. This evaluation assesses ecosystem value on the purpose to promote ecosystem health and the adoption of EbA policies. This study expands awareness of the benefits of ecosystem services under the growing threat of climate change at the study site. In addition, this study acknowledges that the market economy is often misled by the consistent pursuit of growth, and multiple problems have arisen as a result of this.

The effects of climate change alter the distributions of temperature and rainfall; such changes coupled with human exploitation of hillside resources lead to more frequent floods and landslides, resulting in the fragmentation of hillside ecosystems. Without protection, hillside areas gradually lose their ecosystem services. In contrast to destructive growth that threatens communities, sustainable growth is exemplified by local, prosperous ecovillages that create a balance between nature and humans [78–81]. As an ecological economist, Daly [82] proposed that the planet is now full of human populations and human activities, and the balance between man and nature should be pursued.

### 3.6. EbA Practices to Benefit Local Resident

In the study site, the hillside forestry is managed by close cooperation of the government and the local community. Taiwan's government has implemented a policy of Community Forestry for years [83]. In addition, the local residents are involved and empowered. Although the term, EbA, is not used directly in the watershed management practice, the concept of EbA is embedded in existing policies.

Estimations of economic value can be measured from multiple dimensions and can provide references for various aspects of policies. The estimated value of hillside EAS in the present study can supplement the evaluation by Chen et al. [25], who investigated residents' willingness to pay for four categories of ecosystem services, as identified by the MEA [26].

As indicated by the evaluated benefits of EAS, the results demonstrate that humans can live in abundance by protecting ecosystem services, maintaining natural capital, and connecting natural systems with communities. Natural capital can facilitate climate adaptation and provide ecosystem services.

This study analyzed the role and value of ecosystem services under climate change and intensive development. Governments should develop land-use zoning and environ-

mental conservation policies based on land characteristics, and local strategies should be implemented for climate adaptation services. The implementation of EbA practices in communities should encourage community participation and the development of ecotourism and sustainable agriculture [84–89].

### 3.7. Taiwan's Practices and Policy Implications for UNEP

EAS is the ecosystem services that serve to adapt to climate changes. Governing ecosystems for adaptation services is intrinsic to EbA policy. It is the maintenance of ecosystems for emergence of ecosystem services to climate adaptation by increasing the capability of ecosystems to moderate and adapt to climate change and variability. Under increasing climate risks, the EbA policy is advocated by UNEP that finance projects on climate change adaptation all over the world, especially aiming to restore and maintain ecosystems and to benefit local people.

Based on Taiwan's topography of high mountains and steep slopes, a sound soil and water conservation policy is requisite to Taiwan's development and has become the prominent self-financed EbA practices for EAS. Taiwan's EbA policy is fulfilled by EbA practices. The Taiwan government began to develop forward-looking soil and water conservation policies in the early stages of economic development, compositely covering land classification, land use planning, relevant laws and regulations, and the establishment of a dedicated agency system to be responsible for practical measures. Taiwan's government is still working hard to shoulder the works. Now that climate change has become an emergency, Taiwan's long self-financed EbA for EAS would be a paradigm case to the UNEP's global EbA program.

What are the locally viable eco-based practices that can help good ecosystem governance, which is definitely not identical to local communities all over the world? How to design and implement a feasible measure is based on the local natural and social characteristics. Although the various ecosystems would vary, the prudent practices with correspondence would benefit identically.

## 4. Conclusions

Lanyang River watershed is located in northeastern Taiwan. It is a climate-fragile watershed. The local residents would suffer immediately as a result of a climate crisis, such as flooding and landslide, and directly benefit from well-implemented adaptation measures. Local residents are the primary affected group. This study evaluates the benefit value of ecosystem adaptation services (EAS) of a hillside ecosystem supported by eco-based adaptation (EbA) practices in Lanyang River watershed. The single-bounded contingent evaluation method with a logit model was implemented to estimate the economic value for the benefit perceived by the residents. The data are collected by a questionnaire survey with in-person interviews to the resident.

The results reveal that a substantial economic benefit is conferred to the resident from EAS. These quantitative results demonstrate the contributions of ecosystems to climate change adaptation; furthermore, they validate that maintaining and promoting healthy ecosystems to provide climate services constitutes a fundamental climate policy.

The research gives evidence to affirm that the demographic and motivation variables would affect the willingness-to-pay, as proposed by the literature [71].

The perceived threat to life is a factor of significance in the logit regression that reveals a higher tendency to paying for the hypothetical EbA foundation. That the motivation factor, indexed by perceived impacts, would be of importance is evidenced by the finding that the residents perceived higher impacts of climate change would have higher payment tendency.

Residents who are housewives and residents aged 30–60 years old have a higher tendency to pay for the bidding value assigned in the survey. The results evidenced that the housewife could play a role in EbA policy.

It is interesting that the residents of higher income and higher education have higher willingness to pay the assigned bidding amount in the survey. Income is an affordability

indicator; the public self-protection would be limited by income constraints, and government financing in the poor communities would be critical. Moreover, education may increase the corresponding knowledge of residents; education would be a foci element to a biotic measure.

The impacts are serious threats to the life and property of the residents, due to their early access to land reclamation and settlement in the watershed. The inhabitants are under protection, and they are well aware of the impacts and the EAS benefits of EbA measures. Triggered by topographic properties and climate characteristics, Taiwan's composite long-developed EbA policy contributes to the maintenance and emergence of EAS by the local residents. The case of Lanyang watershed and Taiwan EbA would demonstrate a successful example to the United Nations Environmental Program financial and technical assistance in EbA policy.

The results reveal that a substantial economic benefit is conferred to the resident from EAS. Apparently, the evidence offered by this research affirms the prominence of nature-based solutions and eco-based adaptions. The evidence aids with the provision of nature-based solutions to address biodiversity and ecosystem services challenges, with the goal of reducing climate change vulnerability. EBA practices can enhance climate resilience by enabling the power of natural ecosystems to be harnessed to benefit the communities that depend on them. Taiwan's practices can be used as a reference for EbA program of the organization of United Nation Environmental Program. Since the ecosystems would vary with local characteristics, the local viable practices and the benefits would not identical. Future research is suggested on what are the local viable EbA practices, as well as the benefit correspondingly. The demographic variables and motivation variables indexed by perceived impacts are influencing factors to the resident willingness-to-pay. Future research is suggested on the policy implication on these significant variables in global EbA policy application.

**Author Contributions:** Conceptualization, methodology, investigation, software, validation, and formal analysis, writing, and editing: W.-J.C.; organizing research project, survey design, paper review, and paper submission: S.-C.L.; research project administration, W.-J.C., J.-F.J., C.-H.C. and S.-C.L. All authors have read and agreed to the published version of the manuscript.

**Funding:** This research was funded by Ministry of Science and Technology, Taiwan, grant number MOST 111-2321-B-004-001.

**Institutional Review Board Statement:** Not applicable.

**Informed Consent Statement:** Not applicable.

**Data Availability Statement:** Not applicable.

**Acknowledgments:** We gratefully acknowledge five reviewers for their helpful comments and suggestion. The interviewers of this in-person survey are: W.J. Chen; W.R. Chen, C.C. Cheng, I.J. Huang, Y.S. Hua, and W.M. Hsieh.

**Conflicts of Interest:** The authors declare no conflict of interest.

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
