# Peer review of "Do Eco-Based Adaptation Measures Enhance Ecosystem Adaptation Services? Economic Evidence from a Study of Hillside Forests in a Fragile Watershed in Northeastern Taiwan"

_sustainability, doi:10.3390/su15129685_

Round 1
Reviewer 1 Report
Attached

Author Response
The authors wish to thank the anonymous reviewer for his/her useful suggestions regarding the present paper. All comments were addressed in the revised manuscript, along with some major modifications and improvements by the authors, and they have been shown there using the ‘TRACK CHANGES’ WORD property.

Reviewer 2 Report
What is the relevance of Figure 1? It does not explain anything about the analysis except for the repetition of the process already mentioned in the text. Also, the figure is not cross-referenced in the text.
I suggest authors restructure the Literature review and avoid repetition of the same contents again and again. For example, lines 175-176 and 183-184, and 204-206 have similar content.
The question (lines 294-297) designed for the WTP survey is complicated. It is assumed that the ecosystem services/benefits the local community is getting are known to them. Similarly, the impacts of climate change the study area is experiencing are not captured. How is it expected that everyone would know the climate change impacts if not explained?
What EBA policies are is never defined in the paper.
I could not get any specific reference to ecological vulnerability in this paper. Though at many places, authors have mentioned the ‘Ecological Vulnerability’ of Rivers but never came to know about the exposure of the case study area under study. Authors must contextualize the study to strengthen the arguments they are presenting rather than presenting the generic statements already known to all.
Section 4.2 starts with the following statement:
“The present economic valuation analysis reveals the benefit of ecosystem adaptation services related to mitigating the effects of local climate change.”
I didn’t get what exactly are the benefits.
Author Response

(The authors gave the same response as above.)

Reviewer 3 Report
General
The paper deals with interesting topic on ecosystem-based adaptation of hillside in Watershed in Northeastern Taiwan. Nevertheless, my general impression is that many elements remain poorly explained or argued and the global design and its objectives are not clear.
The following are a mix of both minor and more substantial issues that should be addressed.
Comments
1. The introduction is too general. The authors describe a too long relationship between the issues of climate change and the EbA, but in an overly generic or even institutional approach. The introduction does not present the research question or the objectives of the work, nor the method and approach (the choice of an economic evaluation of the contingent valuation type is not neutral), nor the conclusions or the potential impacts in terms of public policy. The specific issue around this case study and the socio-climatic context deserves to be developed so that readers understand this paper.
2. The definition and the differences between ecosystem services and EbA should be clearly presented. Does the study focus on the benefits associated with ecosystem services or on public policy based ecosystems actions?
3. Figure 1 needs to be completely revised, it is confusing and does not add any information to the article as it stands. It is not explained in the manuscript, the boxes and their links are not developed. In addition, several typos need to be corrected: "The benefit of ecosystem-based adaptation" and "Ecosystem-based Adaptation (EbA)”
4. The literature review should be shortened. Section 2.1 should be merged with the first four paragraphs of the introduction to avoid repetition and generalizations about the topic. The literature review should add more case-specific and empirical references to environmental assessments related to the research question.
5. Your contingent valuation do not convince me at all. The evaluation method, its implementation and the survey (the latter included only one question presented on page 7?) are only slightly developed. The scenario is presented too briefly to respondents. Asking individuals about their WTP for the restoration of ecosystem services without any detail is too broad a question that implies many statistical biases (informational, hypothetical, etc.) and strongly reduce your results robustness. Furthermore, the different amount options do not seem to be associated with any real policy or planning action, and seem to be disconnected from a cost-benefit analysis.
6. Explain in detail the different parts and questions of this survey seems necessary? Concretely, what does the vector X contain?
7. In conclusion the authors mention "This study analyzed the direct use value of the ecosystem services to the residents from the perspective of climate adaptation and on the basis of the consumer theory of neoclassical economics" but how can we talk about direct use when we don't know at all what scenario we are talking about. It is not correct to conclude about the measure the benefits of hillside through this study.
Minor comments
- Edit Chen et al. [25] page 3, line 110
- In table 2, put the stars (at 1%, 5% and 10%) to make the reading of the significance levels easier.
Author Response

(The authors gave the same response as above.)

Reviewer 4 Report
This study analyses the “Economic Benefit of Hillside Ecosystem Adaptation Services in a Fragile Watershed in Northeastern Taiwan”. The topic is of high interest, especially in the lack of data for that region. The presentation is clear, the data and statistics robust, and the Discussion section is well structured and derives logically from the Results obtained. I think this work could be published in sustainability subject to minor modifications and clarifications. In addition, I think the study and manuscript fits the journal scope and provides new information.
- My comment refers to the discussion part, which I think could be improved by highliting some parts where comparison with other studies are being done and highlighting community structure of similar ecosystems.
- I also see some grammatical issues in the manuscript which I recommend to double check by authors.
- The first paragraph of the Discussion should give some general view of the work done. With this, I highly recommend changing the order of paragraphs to avoid mixing up the above mentioned subjects.
- I would recommend to have a comprehensive conclusion at the end of Discussion which could be easily driven from what authors obtained.
Author Response

(The authors gave the same response as above.)

Reviewer 5 Report
General comments:
The paper employed a single-bounded contingent valuation method to estimate the willingness to pay of respondents for ecosystem adaptation services, thereby evaluating the economic value of such services. The study found that the average willingness to pay per respondent was 793.65 NT$, which confirms the residents' dependence on the protection of the natural mountain slope ecosystem. The overall structure of the paper is coherent, lucid, and conforms to academic standards. However, there are still some critical issues that require further clarification by the author, as detailed below.
Specific comments:
1.The core issue at hand is the difference between payment for ecosystem services and payment for ecosystem adaptation services. Within your team's existing literature, there has been a discussion of residents' willingness to pay for ecosystem services in mountain forests, with a binary question designed as follows: "With the knowledge that the present ecosystem services will be destroyed one day, would you be willing to pay $ Ai0 to conserve your current ecosystem services now?" The purpose of this question was to evaluate residents' willingness to pay for the conservation of existing ecosystem services provided by nearby forests. In this paper, you have designed a binary question as follows: "Are you willing to pay (NT$) BID per year to enable the foundation to implement the aforementioned policies for climate resilience and adaptation in the hillside ecosystem?" I believe that the two questions are essentially identical.
2. It is important to determine whether the same method can be used to calculate the economic value of ecosystem services and the economic value of ecosystem adaptation services. Previous studies have mostly used contingent valuation methods to estimate the economic value of ecosystem services, and the author has also explained the application of contingent valuation methods in the measurement of the economic value of ecosystem services in the literature review. However, there is a need to supplement the relevant literature on the application of this method in the measurement of ecosystem adaptation services, specifically EbA. Additionally, the literature review needs to be logically linked to the research at hand, with a focus on EbA practice and related literature on economic valuation.
3. The research model requires further elaboration and refinement. On page 7, line 304, the regression model used in this study is briefly described, indicating that the model includes the dependent variable and demographic characteristics of respondents. However, in lines 314-316, it is mentioned that "the significant variables in the logit regression model were retained," and "The significant variables that affected WTP included BID, demographic characteristics, and motivational factors." The author should clearly state which variables were included in the regression model, which variables were removed due to insignificance, and explain the methods used for variable selection.
4. In the results analysis section, there is a need for significant adjustments. (1) In section 4.1, the analysis of regression results lacks a deeper analysis of the factors affecting the results, and only briefly touches upon the topic. (2) In the same section, there is no detailed explanation or analysis of the estimation of the economic value of ecosystem adaptation services, with only a numerical value given. (3) In section 4.2, there is a lack of data to support the discussion of EbA's impact. (4) In section 4.3, there is a similar discussion of EbA's impact, with no connection to the previous calculations. (5) Sections 4.4-4.6 lack any substantive analysis or explanation of results, only emphasizing the importance of EbA valuation, which should be addressed in the introduction section. (6) In section 4.6, there is a mention of the role of private enterprises and NGOs, which has not been previously discussed, and therefore requires clarification.
Author Response

(The authors gave the same response as above.)

Round 2
Reviewer 2 Report
I have read the authors' corrections carefully. The authors have improved the text in the manuscript, especially regarding the inclusion of local context. I appreciate the change you made. There are still some minor issues, like repetition of the same contents, such as lines 52 to 70. You can still write it concisely, which will reduce the word count also. In Lines 115, 124: What does it [1,2525] mean?? Give correct citations. Check the text thoroughly for such typo errors and repetitions in the text.
Author Response

(The authors gave the same response as above.)

Reviewer 3 Report
Overall, the authors have done a nice job addressing the concerns that I expressed in the initial manuscript. Nevertheless, there are still a few major comments.
1. You explain: "According to the literature, the contingent valuation method constitutes a sound method for assessing climate adaptation policies". This methodological justification still seems too weak. Why did you use the contingent valuation method when the Choice Experiment method is the prevailing method for evaluating hypothetical multi-attribute scenarios?
2. Given the importance of the monetary variable in this type of evaluation, I must repeat my question about how the different bidding amounts were defined. The anchoring bias has to be tested.
3. In the section on WTP, you talk about "aforementioned policies", but these are not detailed in the article. Your assessment cannot be understood without explaining the scenarios proposed to respondents.
4. Some of your variables seem to show that your sample is not representative. Why is the "occupational status" limited to only "housewife" status? For the "age" variable, why are there no respondents under the age of 30? To better understand individual preferences, it would have been desirable to see more occupational status and age categories. By limiting the categories, you're making the assumption that non-housewives or 30-60 year-olds have homogeneous preferences, whereas these categories are very broad and therefore heterogeneous.
5. Demographic variables 6, 7 and 8 have disappeared from the regression table. It seems necessary to include them to understand the full effects.
6. Minor remarks :
- To improve readability, move table 2 to the associated section (3.1) and move the stars to the coefficients column.
- L332 edit by (2)age
- L347 edit by parts
Author Response

(The authors gave the same response as above.)

Round 3
Reviewer 3 Report
Thank you and congratulations on the improvements made to your article.